# Heart Rate Response, Temporal Structure and the Stroke Technique Distribution in Table Tennis National Category Matches

**DOI:** 10.3390/ijerph20010739

**Published:** 2022-12-31

**Authors:** Jon Mikel Picabea, Jesús Cámara, Javier Yanci

**Affiliations:** 1Department of Physical Education and Sports, Faculty of Education and Sport, University of the Basque Country (UPV/EHU), 01007 Vitoria-Gasteiz, Spain; 2Society, Sport and Physical Activity (GIKAFIT) Research Group, Department of Physical Education and Sports, Faculty of Education and Sport, University of the Basque Country (UPV/EHU), Lasarteko Bidea 71, 01007 Vitoria-Gasteiz, Spain

**Keywords:** racket sport, game density, cardiac response, ball placement, individual competition

## Abstract

The aims of this study were to analyze the heart rate response, the game temporal structure (i.e., mean total time of the matches, real playing time, total rest time) and the stroke technique distribution and to describe its relations in the table tennis national category in simulated competitions. A cohort of 60 table tennis male players (22.06 ± 8.32 years) played 30 simulated matches. The obtained results show a mean heart rate (HR_mean_) of 142.69 ± 14.10 bpm and a peak heart rate (HR_peak_) of 167.26 ± 16.51 bpm. Total strokes were 7505, being the 57.88% and 42.12% forehand and backhand strokes, respectively. The most frequent forehand strokes were service (SERV) (33.13%) and forward spin technique (SPIN) (40.75%) stroke types, with the left quadrant of the table away from the net (Z_5) (25%) and right quadrant of the table away from the net (Z_6) (20.72%) being the most frequent ball bouncing placements. Meanwhile, the most frequent backhand strokes were backspin technique (PUSH) (42.74%) and SPIN (31.86%) stroke types, with the intermediate left quadrant of the table (Z_3) (17.21%), Z_5 (31.22%) and Z_6 (18.54%) being the most frequent ball placements. The mean total time of the matches was 15.74 ± 3.82 min, the mean real playing time was 4.14 ± 1.47 min and the total rest time was 11.60 ± 2.67 min. Heart rate variables did not correlate significantly with the different stroke types or the game temporal variables (*p* > 0.05). However, stroke types and game temporal structure variables were significantly correlated (*p* < 0.01). This information can be useful to reveal players’ strengths and weaknesses and prepare subsequent training sessions, adapting training sessions to the needs of the athletes.

## 1. Introduction

Table tennis is a worldwide known Olympic sport, with more than 40 million competitive players and about 300 million practitioners around the world [1,2,3]. Due to the large number of practitioners, it is important to know the health status of the table tennis players, especially in amateur category players who usually cannot count on being monitored by qualified personnel in training sessions and competitions [3]. Concretely, the benefits of playing table tennis are related to better bone development and improvements in physical fitness (strength, range of movement and cardiovascular fitness, among others) [2,3], as well as having been recommended as a tool for increasing physical activity [3]. This sport has suffered many rule changes in the last years, such as the inclusion of a plastic ball, increased ball diameter and weight, a new point scoring system to 11 points and prohibition of speed glue or limitations on service technical aspects, among others [4]. These modifications have brought many changes in the game’s structure, but also in heart rate responses [5].

From the point of view of the analysis of the game itself, the players who compete in this sport are required to hit a high-speed ball (>50 km·h^−1^) [6] over 30 times per min during rallies no longer than 4 s with resting times shorter than 15 s [7]. Due to that, high levels of agility, speed, reaction time, coordination, strength and flexibility are essential to perform the required techniques correctly [6,7]. In order to study the required physical capacities of table tennis players, many investigations have been carried out in different situations [6,7,8,9]. These studies have observed differences in some physical capacities (strength, flexibility, among others) according to sex, age, anthropometric attributes and ranking, concluding that the information about how players respond to particular tests helps physical trainers and coaches in setting benchmarks and developing effective training drills accordingly [3]. Nevertheless, the capacity of a table tennis player to win a competition or not is restricted by many factors, such as physical fitness, physiological state, environmental factors and technical and tactical abilities, among others [7,10].

As well as players’ physical condition, one of the most studied factors in racket sports in order to obtain good references for a specific training prescription is the game’s temporal structure. It has been widely studied previously in sports such as badminton [11], tennis [12] and table tennis [2,4,13,14]. Especially, in table tennis, previous studies determined that total match duration ranges from 8 to 38 min for a masculine competition and from 9 to 41 min for a female competition [2,7,13,14]. In addition, Zagatto et al. [7] and Zagatto et al. [13] found similar results in male official competition and simulated matches, respectively, for rally duration (3.5 s), rest time between rallies (8.2 s) and effort and rest ratio of the match (0.44). However, Pradas et al. [2] found in elite individual simulated matches that total match time (2256.05 ± 979.7 s vs. 1469.4 ± 544.2 s) and total resting time (1860.9 ± 838.2 vs. 1104.4 ± 459.1 s) were longer for men, but game density was longer for women (0.27 ± 0.05 vs. 0.41 ± 0.13). Although the game’s temporal structure has been analyzed in table tennis high-level players, it is not known whether the results with players and matches at a lower competitive level may be similar or different. Analyzing the game’s temporal structure in lower competitive level categories could be necessary to better understand the game and to be able to adapt the training and preparation plans according to the demands of the game.

Furthermore, the cardiac demands in table tennis by means of the heart rate (HR) quantification have been previously widely investigated to quantify the physiological demand that players suffer during matches [1,14,15,16]. Previous studies observed heart rate mean values (HR_mean_) of 135–163 beats·min^−1^ [14] or 136–147 beats·min^−1^ [15]. Additionally, it has been shown that HR_mean_ ranges from 68% to 92% of the maximun HR (HR_max_), reporting submaximal intermittent efforts due to table tennis characteristics [7]. Despite previous studies that have analyzed table tennis cardiac responses in national categories, comparing heart rate variability values before and after a simulated competitive match [16], we could not find studies analyzing the HR_mean_ and the HR_peak_ in amateur table tennis players.

In addition to the importance of knowing the heart rate response and the temporal game structure of the matches, table tennis research has also focused on the analysis of technical and tactical variables, using validated notational evaluation methods [10,17,18]. This type of analysis of technical and tactical variables allows for understanding, in depth, other relevant aspects of the game/match. As aforementioned, the technical and tactical ability of the players also often determine the result of the match, and this is reflected in the shot characteristics and concretely in the stroke type, stroke position, stroke efficacy and landing area of the ball [10]. Previous studies that used notational analysis in high-level table tennis players showed that the most used technique was the topspin. Specifically, Wang et al. [10] found that the most used technique in elite women players was the topspin (61.7%) in Olympic Games, whilst Malagoli Lanzoni et al. [17] observed that the most used technique in elite male players was the forehand topspin (27%), the middle half zone of the table being the most frequent ball placement zone in both studies. However, considering that the skill and performance of players at different levels might differ, more studies are needed on lower-level players and matches to better understand the game.

As far as we are concerned, no study has analyzed the correlations between game temporal structure, stroke technique distribution and cardiac demands in table tennis matches. Therefore, the aims of the present study were to analyze the heart rate responses, the game’s temporal structure, the stroke technique distribution and its relations in table tennis national category simulated competitions.

## 2. Materials and Methods

### 2.1. Participants

Sixty male table tennis players (22.06 ± 8.32 yr, 1.73 ± 0.08 m, 63.87 ± 12.75 kg and 21.31 ± 4.15 kg·m^−2^), who were competing during the study period in the Spanish second national league, participated in this study. Inclusion criteria were as follows: (i) at least two years of competitive experience on national level and (ii) must be training on a weekly basis (at least twice a week). Exclusion criteria were: (i) previous injuries that might interfere with the study and (ii) taking medications. Written informed consent was obtained from the players and the club prior to the commencement of the study after receipt of a detailed written and oral explanation of the potential risks and benefits resulting from their participation. Written informed parental consent and player assent were obtained when players were under 18 years of age. Ethical approval was granted by the Ethics Committee for Research on Humans (CEISH, Nº 2080310018-INB0059), and the study was conducted in accordance with the Declaration of Helsinki (2013).

### 2.2. Procedures

In this study, 30 matches played by the 60 male players participating in the study were recorded and analyzed during the off-season period. All players competed during the study period in the Spanish Second National League. Due to the difficulties of analyzing heart rate parameters during official table tennis matches, a simulated competition was designed in accordance with the International Table Tennis Federation (ITTF) rules. All matches were played to the best of five sets. Before each match, a standardized warm-up was performed consisting of 2 min of forehand and backhand rallies as well as forehand and backhand topspin rallies, just as in an official match. Participants were instructed not to exercise vigorously 48 h before the test to avoid any fatigue effects on the results. The matches were recorded using a Samsung Galaxy S10+ mobile camera (Samsung, Suwon, Korea), which was set up on telescopic support (67″ selfie stick, Aureday, Shanghai, China) in the right corner of the table tennis tables at a distance of 5 m and a height of 2.5 m, in order to know the temporal structure and make the observational analysis of the game (Figure 1). The camera recorded both players (Figure 2). In addition, the HR of the players was measured (Polar V800, Kempele, Finland) throughout the match.

### 2.3. Measures

Heart rate analysis: A portable HR monitor (Polar V800, Kempele, Finland) was used to record HR data every second. The data obtained were transferred to the computer via Polar Software V4 (Polar Flow, Kempele, Finland). The following variables were obtained: (i) mean heart rate (HR_mean_), (ii) peak heart rate (HR_peak_) and (iii) minimum heart rate (HR_min_).

Temporal structure: As in a previous investigation, through observational analysis of the matches [2], total time (TT; full match time, from the beginning to the end, considering game and rest periods), total resting time (TRT; sum of the periods during which the ball was not played) and real playing time (RPT; total time, less the TRT) were measured during each match. Apart from these variables, game density (playing/resting times) was also analyzed [2].

Observational analysis tool: A previously validated observational tool was used to carry out the analysis [18]. The observational instrument was incorporated into Lince PLUS software (version 1.3.2, Spain), which is freely available for observational analysis [19] (Figure 2). The videos were played at ×0.25 of the speed with the Lince PLUS software to precisely determine the ball placement. The annotation system, with the criteria and categories established, is shown in Table 1.

### 2.4. Statistical Analysis

The results are presented as means ± standard deviation (SD). The observational analysis data are presented as frequencies and percentage (%). Data normality was evaluated with a Shapiro–Wilk test, which determined that the assumption of normality was violated for the game’s temporal structures of variables and stroke types (*p* < 0.05). The chi-square test was used to analyze the differences in the percentage distributions. Spearman’s rank correlation analysis (Rho) was used to determine the relationships between different variables groups (game temporal structure, heart rate recording and the stroke types). The magnitude of correlation between tests was assessed with the following thresholds: <0.1, trivial; =0.1–0.29, small; <0.3–0.49, moderate; <0.5–0.69, large; <0.7–0.89, very large and <0.9–1.0, almost perfect [3]. The data analysis was carried out using the Statistical Package for Social Sciences (SPSS Inc., version 25.0, Chicago, IL, USA).

## 3. Results

Game temporal structure analysis results and the HR data analysis during the simulated matches are shown in Table 2. The HR_mean_ of the 60 players that participated in the 30 matches of the study was 142.69 ± 14.10 bpm. The 29.97% of the matches included 3 sets, 26.64% included 4 sets and 43.29% included 5 sets. In addition, the mean total time of the matches was 15.74 ± 3.82 min, the mean real playing time was 4.14 ± 1.47 min and the total rest time was 11.60 ± 2.67 min.

Table 3 shows the stroke distribution during the simulated competition divided into forehand and backhand strokes. Total strokes were 7505, being the 57.88% and 42.12% forehand and backhand strokes, respectively. The most frequent forehand strokes were SERV (33.13%) and SPIN (40.75%) stroke types, with the Z_5 (25%) and Z_6 (20.72%) being the most frequent ball bouncing placements. Meanwhile, the most frequent backhand strokes were PUSH (42.74%) and SPIN (31.86%) stroke types, with Z_3 (17.21%), Z_5 (31.22%) and Z_6 (18.54%) being the most frequent ball placements. Looking to the techniques used by the players depending on the stroke side, there are forehand SERV and SPIN strokes and backhand PUSH stroke predominance (*p* < 0.01) (Figure 3). In addition, looking to the technique used and the ball placements, SERV technique is usually directed to Z_3, PUSH techniques are usually directed to the Z_3, Z_4 and Z_5 zones, NORM techniques are usually directed to the Z_5 zone and SPIN strokes are directed to Z_5 and Z_6 (*p* < 0.01) (Figure 4).

Regarding correlations, the HR variables did not correlate significantly with the different stroke types or the game temporal variables (*p* > 0.05). However, stroke types and game temporal structure variables correlated significantly. Concretely, SERV, NORM and PUSH strokes correlated significantly with TT (r = 0.51 to 0.77, moderate to large, *p* < 0.01), RPT (r = 0.48 to 0.76, moderate to large, *p* < 0.01) and TRT (r = 0.50 to 0.67, moderate, *p* < 0.01) (Table 4).

## 4. Discussion

The purposes of this study were to analyze the heart rate response, the game’s temporal structure and the stroke technique, distribution and ball placement in table tennis simulated competitions. Even though previous studies have analyzed the heart rate response [2,20,21,22,23], the game structure [2,22,23], the stroke technique and ball placement [10,24,25,26] in table tennis, the strength of this study lies in the fact that the data are obtained from national categories and the analysis of correlations between the aforementioned variables. Therefore, these results might be more useful to coaches training lower category groups and players themselves to adequate training sessions.

Cardiac response is a commonly used parameter in table tennis to characterize the players’ physiological demands during matches [2,4,21,27]. We observed that the match HR_mean_ was 142.69 ± 14.10 bpm and HR_peak_ was 167.26 ± 16.51 bpm. These results come in accordance with previous studies that reported a HR_mean_ between 137 and 176 bpm for male competitions [2,20,27]. HR_peak_ has been observed to range between 160 and 180 bpm in official and simulated male competitions [2,20,21,25]. It must be considered that HR_mean_ and HR_peak_ could be affected by many factors, such as the used material (i.e., sport material and ball size, slowing todays table tennis playing), the specific competition situations such as concrete scores or decisive points, game styles (offensive, defensive, mixed) and tactics, or even the gender [2]. These results confirm the results obtained in previous studies, showing that table tennis is an intermittent sport due to effort/rest ratio, but high HR values are obtained during rallies because of the speed and accuracy of the strokes and movements [7].

Regarding game temporal structure, the results of the present study are consistent with the results of previous studies. It has been observed in different level categories (regional, national and Olympic) that competition TT can last between 8 and 38 min [7,14,15,27]. Nevertheless, since set number varies across categories, game temporal structure might differ across categories. Specifically, Pradas et al. [2] reported the TT of men’s and women’s categories. It was observed that men’s matches were longer than women’s matches (37 min and 24 min of TT, respectively). In this line, Kasai et al. [20] reported that top world player matches can last up to 45 min. These previous studies observed higher TT than those obtained in this study (from 235.07 to 285.89%). Variations in the game’s temporal structure might be due to differences in competitors’ level, competition phase or players’ game style [2,4,7]. In addition, the matches of this study were played to the best of five sets, while the previous studies matched were played to the best of seven sets, following the International Table Tennis Federation (ITTF) rules, affecting to the TT results. Another variable used to analyze the game’s temporal structure is the game density [2,4,23], that shows the ratio between the RPT and the TRT variables. Previous studies in table tennis showed game densities between 0.15 and 0.5 [2,4,23], being the results obtained in this study in that range. However, game density is also affected by competitors, competition level, competition phase and the players’ game style (defensive, mixed, offensive) [2]. Moreover, De Mello Leite et al. [4], in a study with 16 elite male players, found game densities from 0.15 to 0.22 in official tournaments. The rest time increased within an ongoing tournament (i.e., quarterfinal matches vs. semifinals vs. finals matches). Respecting simulated matches, Zagatto et al. [13] found a 0.5 game density, while Pradas et al. [2] found 0.27 game density for male categories. These results show that table tennis, also in the category analyzed in this study, is characterized with high rest times due to the time spent for an adequate recovery, preparation and concentration for a new rally [7].

Stroke technique, distribution and ball placement have been previously analyzed in top level table tennis players [10,17,28] reporting different results. Additionally, Wang et al. [10] observed that the topspin was the most used technique (61.7%) in the women’s singles finals and semifinals from 2004 to 2021 Olympic Games. In this same study, it was shown that the most used stroke technique was the backhand position (52.4%), followed by the forehand position (29.6%) and, finally, pivot (15.2%). In addition, Malagoli Lanzoni et al. [17] found in elite male players that most of the strokes were forehand topspins (27%), followed by serves (20%), with most of the serves being executed forehand. In addition, Malagoli Lanzoni et al. [28] found, in a 10 player sample from the 30 top male players in the 2008–2010 period, that most of the strokes were the forehand topspins (19.5%), followed by the forehand counter topspins (16.7%) and backhand blocks (14.9%), with a forehand predominance. These results come partially in accordance with those obtained in this study. The present study shows a forehand stroke predominance (57.9%), the forehand topspin (23.6%) and the forehand serve (19.2%) being the most used techniques. Regarding ball placement, Malagoli Lanzoni et al. [17] observed that 52.4% of the serves were performed to the Z_3 and Z_4 zones. In addition, Wang et al. [10] found that the majority of the strokes were performed to the middle half zone of the table (19.27%), corresponding to the Z_3 and Z_4 zones of this study, followed by the middle long zone (16.25%) and backhand long zone (15.20%), corresponding to Z_5 of this study. These results differed with those obtained in the present study, since they observed a Z_5 and Z_6 ball placement predominance in both forehand and backhand strokes. This difference might be due to the players’ level, since top players look for a middle landing area to prevent the opponent from attacking or to limit the fast attack of the opponent’s backhand stroke [11]. These tactical differences could explain the big predominance of forehand topspin strokes compared with the other strokes in this study, due to the players of the studied category being unable to prevent the opponent from attacking as the players of higher-level categories do.

Thus far, no studies have been published that analyze the correlations between HR variables, stroke types and game temporal structure in table tennis. This information may help to understand whether temporal variables and stroke types may influence in HR in table tennis players. In the present study, we observed significant correlations between stroke types and game temporal structure, but no correlation was found between HR variables with stroke types and game temporal structure. The absence of a correlation between HR variables and the other ones can be due to the intermittency that occurs throughout the game, which allows for relative recovery from the used techniques and match duration. As described above, table tennis is an intermittent sport with significant rest times between points [4]. This aspect could partially explain the absence of associations between HR variables with the stroke types or temporal variables. Regarding the correlation between stroke types and the game structure variables, these results may be due to the fact that as the number of strokes increases, the duration of the match will also increase, thus increasing both real playing time and real rest time. In addition, SPIN technique did not correlate with game structure variables, and this may be because the SPIN stroke is the fastest technique in table tennis, so it could explain that slower strokes such as the SERV, NORM and PUSH stroke could correlate with game structure variables. However, as we know, this is the first study that correlates the aforementioned variables, so more studies are needed to compare the obtained results and better understand the obtained associations.

The present study is not without limitations that might have affected the results: (i) the study included only players with an offensive playing style; (ii) only simulated matches were analyzed; (iii) only men’s matches were taken into account; (iv) the distance from the table and the position of the players were not considered, which are factors of paramount importance due to the high tactical and technical demand in table tennis; (v) sport materials (wood and rubber coatings) and specific competition situations, such as scores or decisive points, that can impact HR and technical–tactical aspects, were not taken into account; (vi) only one camera was used and that could have affected the annotation of the ball placement; (vii) there are other observational tools that could provide interesting information [11,29], in addition to that already given. Therefore, future investigations are needed that analyze table tennis players’ physical demands in official matches, including both genders, with a technical and tactical analysis.

## 5. Conclusions

The obtained results show the heart rate response, temporal structure, stroke frequencies and ball placement of male table tennis players. The total time, effort/rest ratio, HR_mean_ and HR_peak_ of simulated matches are similar to the results obtained in previous studies that analyzed the aforementioned variables in official matches, but these could change depending on the category. Regarding realized strokes and ball placement, it looks that the forehand topspin and forehand serves are the most frequent techniques, looking for the farthest quadrants for topspin technique and intermediate quadrants for serves. In addition, the correlation between stroke types and game temporal structure indicates that an increase in the number of slower techniques (i.e., SERV, PUSH and NORM) increases match duration, real played time and real rest time.

## Figures and Tables

**Figure 1 ijerph-20-00739-f001:**
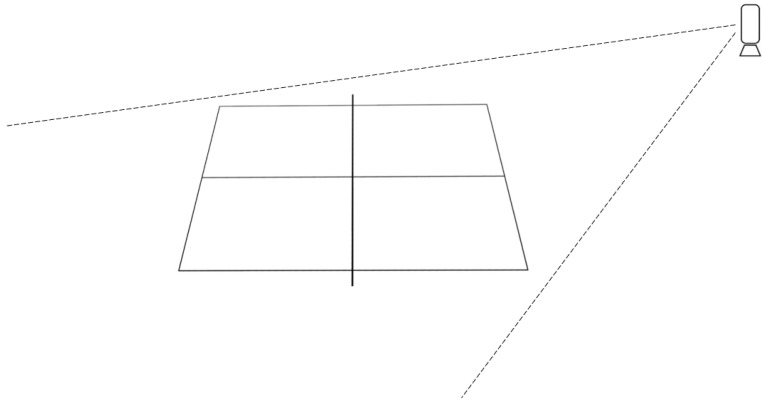
Video-recording protocol.

**Figure 2 ijerph-20-00739-f002:**
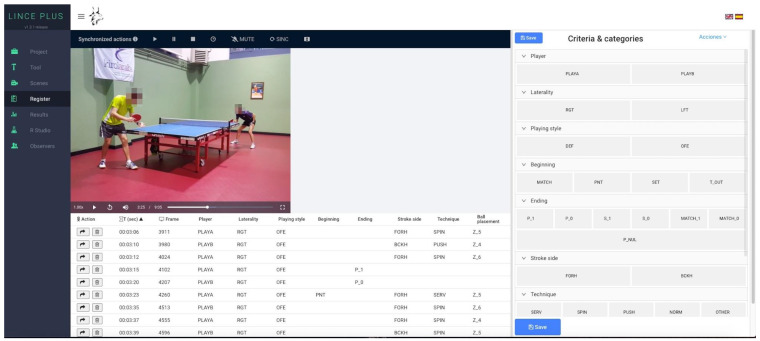
Lince observational tool interface.

**Figure 3 ijerph-20-00739-f003:**
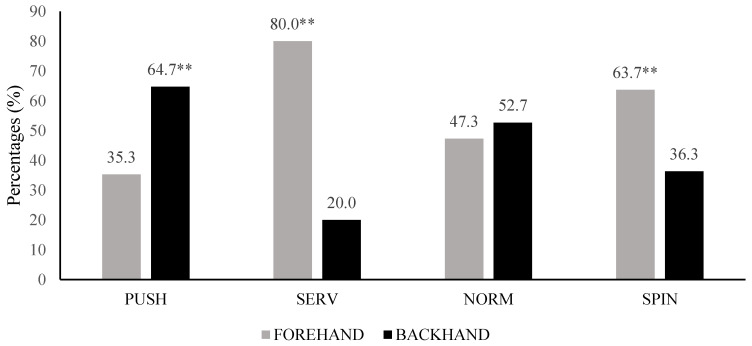
Distribution of the techniques depending on the stroke side. SERV= serve technique; SPIN = forward spin technique; PUSH = backspin technique; NORM = no spin technique; Significant differences (** *p* < 0.01) in predominance of the technique.

**Figure 4 ijerph-20-00739-f004:**
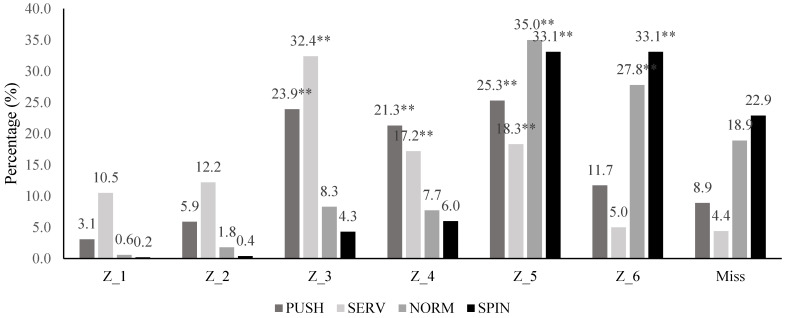
Distribution of the ball placement depending on the used technique. SERV= serve technique; SPIN = forward spin technique; PUSH = backspin technique; NORM = no spin technique; Z_1 = left quadrant near to the net; Z_2 = right quadrant near to the net; Z_3 = intermediate left quadrant; Z_4 = intermediate right quadrant; Z_5 = left quadrant away from the net; Z_6 = right quadrant away from the net. Significant differences (** *p* < 0.01) in predominance of the ball placement for each technique.

**Table 1 ijerph-20-00739-t001:** Codification of the notational system into criteria and stroke type categories.

Player
PLAYA: Player serving
PLAYB: Player receiving
Laterality
RGT: Right-handed player
LFT: Left-handed player
Playing style
DEF: The player uses a defensive playing style
OFE: The player uses an offensive playing style
Beginning
MATCH: The match begins
PNT: The point begins
SET: The set begins
T_OUT: A time-out is requested
Ending
P_1: The player wins the point
P_0: The player loses the point
S_1: The player wins the set
S_0: The player loses the set
MATCH_1: The player wins the match
MATCH_0: The player loses the match
P_NUL: The point is considered as null
Stroke side
FORH: The stroke is done with the forehand sideBCKH: The stroke is done with the backhand side
Technique
SERV: Serve
SPIN: Stroke done close (flip) or far (topspin) from the table where the ball is given a spin effect with a down-up and back-forward racquet movement.
PUSH: Short distance stroke inside the table (push) or away from the table (chop) where the ball is given a backspin effect with an up-down and back-forward racquet movement.
NORM: Stroke done close (block) or medium distance (attack or smash) from the table where no effect is given to the ball with a back-forward racquet movement.
OTHER: Another technique not described in the previous ones
Ball placement
Z_1: The ball bounces in the left quadrant near the net.
Z_2: The ball bounces in the right quadrant near the net.
Z_3: The ball bounces in the intermediate left quadrant.
Z_4: The ball bounces in the intermediate right quadrant.
Z_5: The ball bounces in the left quadrant away from the net.
Z_6: The ball bounces in the right quadrant away from the net.
Side rotation
ROT: Side change in the last set

**Table 2 ijerph-20-00739-t002:** Heart rate responses during the simulated competition and game temporal structure analysis.

	Mean ± SD	Min.	Max.
Heart rate responses			
HR_mean_ (bpm)	142.69 ± 14.10	107.71	171.5
HR_min_ (bpm)	104.81 ± 16.98	68.07	140.42
HR_peak_ (bpm)	167.26 ± 16.51	128.59	202.18
Game temporal structure			
TT (min)	15.74 ± 3.82	7.13	21.63
RPT (min)	4.14 ± 1.47	1.77	7.51
TRT (min)	11.60 ± 2.67	4.33	15.31
GD	0.36 ± 0.12	0.21	0.76

Note: SD = Standard deviation; Min. = minimum values; Max. = maximum values; HR_mean_ = mean heart rate; HR_min_ = minimum heart rate; HR_peak_ = peak heart rate; bpm = beats per minute; TT = total time; RPT = real playing time; TRT = total resting time; GD = game density.

**Table 3 ijerph-20-00739-t003:** Distribution (percentages and frequencies) of stroke type and ball placement throughout the simulated competition.

Forehand Stroke Techniques		Z_1% (*n*)	Z_2% (*n*)	Z_3% (*n*)	Z_4% (*n*)	Z_5% (*n*)	Z_6% (*n*)	Miss% (*n*)	Total% (*n*)
	SERV	3.55% (154)	3.89% (169)	10.64% (462)	5.52% (240)	6.42% (279)	1.61% (70)	1.50% (65)	33.13% (1439)
	SPIN	0.00% (0)	0.23% (10)	1.61% (70)	2.88% (125)	12.48% (542)	14.85% (645)	8.70% (378)	40.75% (1770)
	PUSH	0.78% (34)	1.59% (69)	3.66% (159)	5.52% (179)	3.34% (145)	1.50% (65)	1.96% (85)	16.94% (736)
	NORM	0.12% (5)	0.12% (5)	0.81% (35)	1.04% (45)	2.76% (120)	2.76% (120)	1.59% (69)	9.19% (399)
	OTHER	0.00% (0)	0.00% (0)	0.00% (0)	0.00% (0)	0.00% (0)	0.00% (0)	0.00% (0)	0.00% (0)
	TOTAL	4.44% (193)	5.82% (253)	16.71% (726)	13.56% (589)	25.00% (1086)	20.72% (900)	13.74% (597)	100% (4344)
**Backhand Stroke Technique**		**Z_1% (*n*)**	**Z_2% (*n*)**	**Z_3% (*n*)**	**Z_4% (*n*)**	**Z_5% (*n*)**	**Z_6% (*n*)**	**Miss% (*n*)**	**Total% (*n*)**
	SERV	1.11% (35)	1.58% (50)	3.80% (120)	2.18% (69)	1.58% (50)	0.63% (20)	0.47% (15)	11.36% (359)
	SPIN	0.16% (5)	0.00% (0)	1.58% (50)	1.33% (42)	11.96% (378)	8.64% (273)	8.19% (259)	31.86% (1007)
	PUSH	0.95% (30)	1.71% (54)	10.72% (339)	8.38% (265)	12.15% (384)	5.66% (179)	3.16% (100)	42.74% (1351)
	NORM	0.00% (0)	0.32% (10)	1.11% (35)	0.63% (20)	5.54% (175)	3.61% (114)	2.85% (90)	14.05% (444)
	OTHER	0.00% (0)	0.00% (0)	0.00% (0)	0.00% (0)	0.00% (0)	0.00% (0)	0.00% (0)	0.00% (0)
	TOTAL	2.21% (70)	3.61% (114)	17.21% (544)	12.53% (396)	31.22% (987)	18.54% (586)	14.68% (464)	100% (3161)

Note: SERV= serve technique; SPIN = forward spin technique; PUSH = backspin technique; NORM = no spin technique; OTHER = any other technique; Z_1 = left quadrant near to the net; Z_2 = right quadrant near to the net; Z_3 = intermediate left quadrant; Z_4 = intermediate right quadrant; Z_5 = left quadrant away from the net; Z_6 = right quadrant away from the net.

**Table 4 ijerph-20-00739-t004:** Correlation analysis among stroke type, heart rate and game structure variables for all table tennis players.

	SERV	SPIN	PUSH	NORM	HR_mean_	HR_min_	HR_peak_	TT	RPT	TRT
SERV										
SPIN	0.26									
PUSH	0.47 **	0.04								
NORM	0.49 **	0.14	0.46 **							
HR_mean_	0.02	0.14	0.05	0.18						
HR_min_	−0.05	0.03	0.08	0.10	0.72 **					
HR_peak_	0.00	0.08	−0.04	0.12	0.88 **	0.50 **				
TT	0.59 **	0.07	−0.09	0.09	0.02	0.07	−0.04			
RPT	0.59 **	0.23	0.48 **	0.76 **	0.04	−0.02	−0.09	0.90 **		
TRT	0.57 **	0.16	0.50 **	0.67 **	0.18	0.12	0.11	0.96 **	0.77 **	

Note: SERV= serve technique; SPIN = forward spin technique; PUSH = backspin technique; NORM = no spin technique; HR_mean_ = mean heart rate; HR_min_ = minimum heart rate; HR_peak_ = peak heart rate; bpm = beats per minute; TT = total time; RPT = real playing time; TRT = total resting time. Significant correlation (** *p* < 0.01) between variables.

## Data Availability

The data presented in this study are available on request from the corresponding author. The data are not publicly available due to confidentiality and anonymity of study participants.

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
