# Peer review of "Heart Rate Response, Temporal Structure and the Stroke Technique Distribution in Table Tennis National Category Matches"

_ijerph, 2022, doi:10.3390/ijerph20010739_

Round 1

Reviewer 1 Report

Thank you for your submission. I believe that this manuscript needs some minor revisions, but has the potential to be published in the future.
I have the following suggestions for the authors:

The methods section should be precisely described, with a special emphasis on the missing statistical methodology. Authors should describe the video analysing methods more precisely. For example how the authors determined the exact ball placement. Is there any additional tool in the used software to evaluate the exact place of the ball?
On the other hand, how many observer analysed the recorded video? Did you examine (inter)-observer reliability?
What is the concept of the selection of the analysing categories?

Reviewer 2 Report

I am grateful for the opportunity to review this manuscript titled "Game analysis and match physiological profile of national category table tennis players”. The purpose of this study was to analyze the physiological demands, the game temporal structure, the stroke technique distribution and to describe its relations in table tennis national category in simulated competitions. The data collected in this study may affirm or expand on available literature.

This study is of interest to the IJERPH readers and seems to provide some new findings, applicable to the fields of training. However, the points mentioned below should be considered and the manuscript amended accordingly before being considered for publication.

General comments:

1.      In the presented manuscript, the authors wrote "match physiological profile" in the title; then in the abstract and in the following sections of the manuscript (introduction, methods, results, discussion, conclusion) the authors uses the terms "physiological demands", "physiological response", "cardiac demands", etc. But in all sections of the manuscript, the authors describes, refers to, and measures only heart rate. So it only deals with one of many physiological indicators. Therefore, I suggest that instead of "physiological..." use the term "heart rate"; for example: heart rate observations, heart rate recording, or heart rate response. This comment applies to the entire manuscript, including the title.

2.      The authors studied 60 participants who formed one group. In addition to the analysis of the entire study group, it might have been worthwhile to additionally divide the participants into several groups, e.g. by their place in the national ranking (number of points in the national ranking), fitness level or the number of training sessions performed per week. Such a comparison would certainly be more interesting. This can be considered as a note for the future, or you may be tempted to do this additional analysis during the current revision of the manuscript.

3.      In the presented manuscript, the authors wrote that they measured maximum heart rate (HRmax). However, to measure the maximum heart rate should be done maximum intensity effort. I have doubts about maximum effort during a match, so I suggest using the term peak heart rate (HRpeak). The commentary applies to the entire manuscript.

Specific comments:

Abstract:

1.      The first three sentences in the abstract are unnecessary. They can be removed.

2.      The authors use abbreviations (Z_3; Z_5; Z_6) and did not explain them.

3.      The authors use the terms "game temporal variables" or "game temporal structure" without explaining what they mean. So on a abstract level  it's not clear.

Introduction

1.      In the introduction section, paragraphs 3, 4, and 5 start with the same phrase. Maybe it's worth changing.

Materials and Methods

1.      Point 2.2. Procedures: Line 125-127. Is it really before the match (also an official match) that the warm-up lasts only 2 minutes, and consists only in bouncing the ball ?

2.      Point 2.3. Measures: Line 142-144. Calculation of HRmax based on the formula is very inaccurate. Because the real HRmax (measured during maximum effort) in some athletes is very different from the result obtained according to the calculations from the given formula.

Therefore, I suggest removing this element throughout the manuscript (Methods, Results, and Discussion sections).

Discussion

1.      Line 237: " … with aerobic predominance, … " - For such a statement, the measurement of HR alone is definitely not enough. For such a statement, metabolic measurements should be performed.

Round 2

Reviewer 2 Report

The authors have improved the manuscript according to the recommendations and reaching the good standards.

Please remove the column with HRmaxT in Table 4 which was left over as an oversight.
